# Chondroitin sulfate proteoglycan 4,6 sulfation regulates sympathetic nerve regeneration after myocardial infarction

Matthew R Blake[1], Diana C Parrish[1], Melanie A Staffenson[1], Shanice Sueda[2], William R Woodward[1], Beth A Habecker[1]*

[1]Department of Chemical Physiology and Biochemistry, Oregon Health and Science University, Portland, United States; [2]Portland State University EXITO Scholars Program, Portland State University, Portland, United States

*For correspondence:
habecker@ohsu.edu

Competing interest: The authors declare that no competing interests exist.

**Abstract** Sympathetic denervation of the heart following ischemia/reperfusion induced myocardial infarction (MI) is sustained by chondroitin sulfate proteoglycans (CSPGs) in the cardiac scar. Denervation predicts risk of sudden cardiac death in humans. Blocking CSPG signaling restores sympathetic axon outgrowth into the cardiac scar, decreasing arrhythmia susceptibility. Axon growth inhibition by CSPGs can depend on the sulfation status of the glycosaminoglycan (CS-GAG) side chains. Tandem sulfation of CS-GAGs at the 4th (4S) and 6th (6S) positions of n-acetyl-galactosamine inhibits outgrowth in several types of central neurons, but we don't know if sulfation is similarly critical during peripheral nerve regeneration. We asked if CSPG sulfation prevented sympathetic axon outgrowth after MI. Reducing 4S with the 4-sulfatase enzyme Arylsulfatase-B (ARSB) enhanced outgrowth of dissociated rat sympathetic neurons over CSPGs. Likewise, reducing 4S with ARSB restored axon outgrowth from mouse sympathetic ganglia co-cultured with cardiac scar tissue. We quantified enzymes responsible for adding and removing sulfation, and found that CHST15 (4S dependent 6-sulfotransferase) was upregulated, and ARSB was downregulated after MI. This suggests a mechanism for production and maintenance of sulfated CSPGs in the cardiac scar. We decreased 4S,6S CS-GAGs in vivo by transient siRNA knockdown of *Chst15* after MI, and found that reducing 4S,6S restored tyrosine hydroxylase (TH) positive sympathetic nerve fibers in the cardiac scar. Reinnervation reduced isoproterenol induced arrhythmias. Our results suggest that modulating CSPG-sulfation after MI may be a therapeutic target to promote sympathetic nerve regeneration in the cardiac scar and reduce post-MI cardiac arrhythmias.

## Editor's evaluation

This is an extremely well performed study that advances the knowledge in this area and will foster further work. It also showcases how basic neuroscientific work can be applied to complex pathophysiological problems.

## Introduction

Sympathetic denervation following myocardial infarction (MI) is well documented in both animal models and humans, and highly predictive of ventricular arrhythmias and sudden cardiac death (*Boogers et al., 2010*; *Fallavollita et al., 2014*; *Nishisato et al., 2010*). Following an early period of axon degeneration after MI caused by ischemia/reperfusion (I/R), sympathetic nerves grow back through undamaged myocardium but do not enter the scar due to the presence of chondroitin sulfate proteoglycans (CSPGs) (*Gardner and Habecker, 2013*). This heterogeneity in sympathetic innervation

increases risk for arrhythmias and sudden cardiac death (*Herring et al., 2019*). A therapeutic intervention restoring sympathetic innervation in the cardiac scar by modulating CSPGs may improve health outcomes for patients suffering post-MI arrhythmias.

CSPGs are a diverse family of extracellular matrix proteins modified by chondroitin sulfate (CS) side chains that inhibit nerve regeneration in numerous injury paradigms including MI, spinal cord injury (SCI), and traumatic brain injury (TBI) (*Brown et al., 2012*; *Gardner and Habecker, 2013*; *Lang et al., 2015*; *McKeon et al., 1999*; *Miller and Hsieh-Wilson, 2015*; *Yi et al., 2012*). CSPGs are heterogeneous but all are composed of a core protein that is covalently linked to repeating disaccharide side chains known as glycosaminoglycans (GAGs). While some evidence suggests that the CSPG core protein itself plays a role in regulating axon outgrowth (*Dou and Levine, 1994*; *Ughrin et al., 2003*), the primary effect is thought to occur via the post-translational addition of sulfate to GAGs (*Mencio et al., 2021*; *Miller and Hsieh-Wilson, 2015*). CS-GAGs in scar tissues bind CSPG receptors like PTPσ (protein tyrosine phosphatase receptor sigma) on regenerating axons (*Coles et al., 2011*; *Shen et al., 2009*). CSPG sulfation is attached to specific locations of the CS-GAG by sulfotransferase enzymes, yielding unique structures that differentially affect axon outgrowth. Specifically, 4-sulfation (4S) and 6-sulfation (6S) of *N*-acetyl-galactosamine are critical regulators of axon outgrowth (*Brown et al., 2012*; *Gilbert et al., 2005*; *Wang et al., 2008*). 4S CS-GAGs are produced by the chondroitin-4-sulfotransferase, CHST11, while 4,6-tandem sulfated GAGs (4S,6S) are produced by a 4S-dependent chondroitin-6-sulfotransferase, CHST15 (*Miller and Hsieh-Wilson, 2015*). 4S CS-GAGs can also be removed by an endogenously expressed 4-sulfatase, arylsulfatase-B (ARSB) (*Figure 1A*; *Pearson et al., 2018*; *Wang et al., 2008*). Evidence from SCI and TBI indicate that tandem sulfated 4S,6S CS-GAGs potently suppress axon outgrowth (*Brown et al., 2012*; *Gilbert et al., 2005*). Removal of all CS-GAGs from cardiac scar tissue by treatment with the enzyme chondroitinase ABC (chABC) restores sympathetic axon outgrowth in vitro (*Gardner and Habecker, 2013*) but it remains unknown whether CSPG sulfation is critical to inhibit regeneration of peripheral nerves.

Here, we show that 4,6-tandem sulfated CSPGs are enriched in the cardiac scar after MI, and that this sulfation is important for preventing nerve regeneration in the heart. Sulfation-related enzymes are altered in the heart after MI, and reducing sulfation of CS-GAGs by transient siRNA knockdown of *Chst15* promotes reinnervation of the cardiac scar. Reinnervation decreases isoproterenol-induced arrhythmias.

## Results

### 4,6-Sulfation of CS-GAGs increased in the heart after I/R

To establish whether CSPG sulfation occurs after MI caused by I/R, we used antibodies specific to 4S and 6S CS-GAGs (*Yi et al., 2012*) to compare CSPG sulfation in mouse unoperated left ventricle to day 14 post-I/R scar tissue. Day 14 represents a time point when scar tissue is relatively stable and sympathetic nerves are excluded from entering the scar. I/R led to increased 4,6-sulfation of CSPGs in the cardiac scar (*Figure 1B–D*). In contrast, CSPG levels are low in remote myocardium (*Gardner and Habecker, 2013*), and 4S is not altered after I/R (*Figure 1—figure supplement 1*).

### Reducing 4S with ARSB promotes sympathetic neurite outgrowth in vitro

To test whether 4S of CS-GAGs prevented neurite outgrowth across CSPGs, we enzymatically removed 4S from purified CSPGs using the 4-sulfatase, ARSB. An effective ARSB dose was identified in pilot studies to remove 4S while leaving 6S intact (*Figure 2—figure supplement 1*). Dissociated sympathetic neurons from neonatal rats were then grown on CSPGs treated with vehicle or ARSB. Sympathetic neurite extension across untreated CSPGs was suppressed 40 hr post-plating compared to laminin alone, while ARSB treatment restored neurite outgrowth in a dose-dependent manner (*Figure 2A and B*). These results indicate that 4S of CS-GAGs inhibits sympathetic neurite extension.

### Reducing 4S of cardiac scar tissue with ARSB restores sympathetic axon outgrowth ex vivo

We treated mouse myocardial I/R explants with ARSB to test whether removing 4S from CS-GAGs in cardiac scar tissue could restore sympathetic axon outgrowth in explant co-cultures (*Figure 3A*).

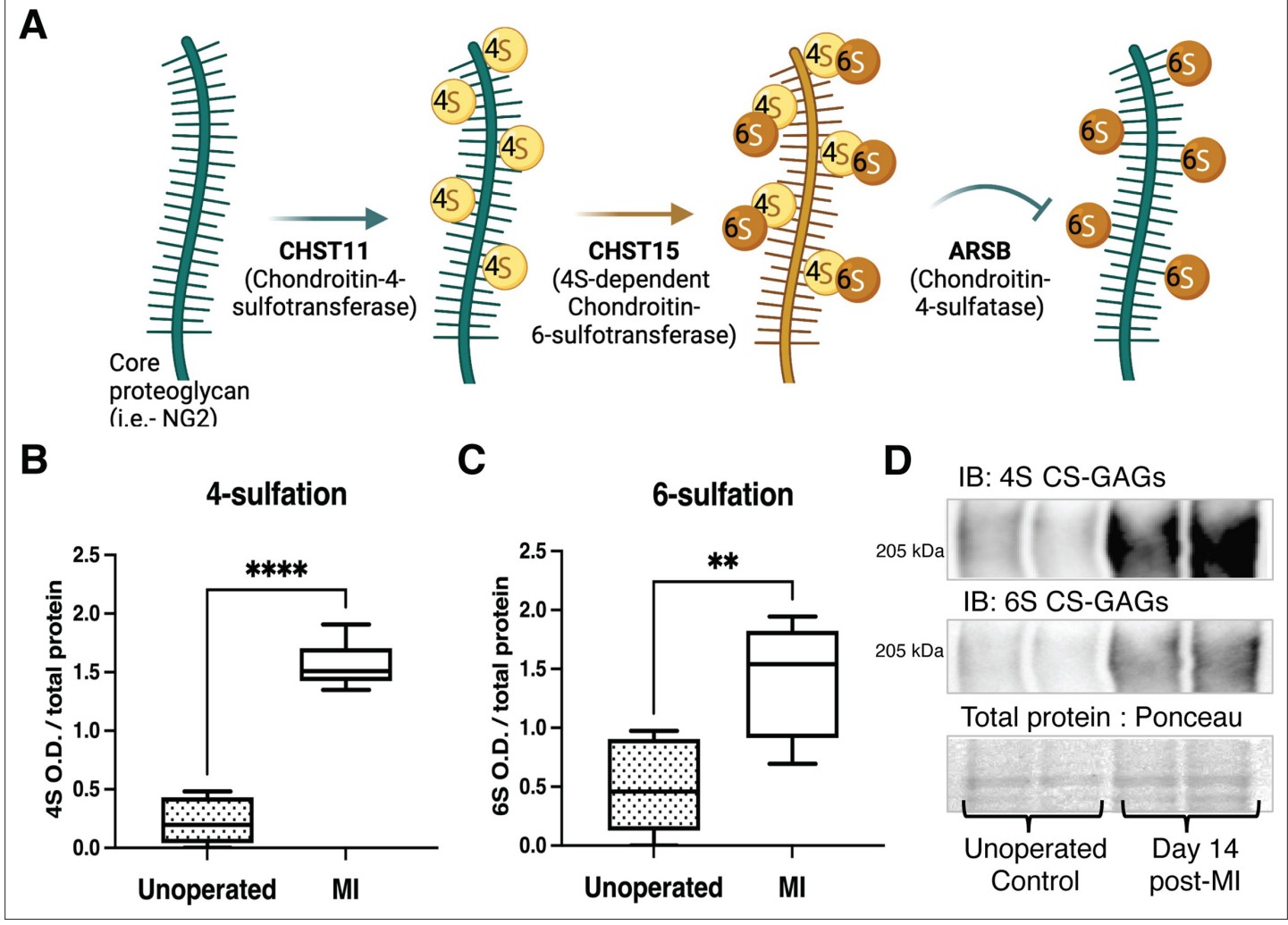

**Figure 1.** Chondroitin sulfate proteoglycan (CSPG) 4,6-sulfation increases in the cardiac scar after myocardial infarction (MI) caused by ischemia/reperfusion (I/R). (**A**) CSPG sulfation patterning schematic with key enzymes. (**B**) 4-Sulfation (4S CS-GAGs) or (**C**) 6-sulfation (6S CS-GAGs) quantification assessed by western blot in the healthy myocardium (unoperated) or cardiac scar 14 days after MI (MI). Quantification of n=6 animals per treatment group, mean optical density (OD) ± SD, Student's t-test (Welch's test), 4S ****p-value < 0 .0001, 6S **p-value = 0.003. (**D**) Example blot images for 4S CS-GAGs, 6S CS-GAGs, and total protein from two unoperated and two MI animals.

The online version of this article includes the following figure supplement(s) for figure 1:

**Figure supplement 1.** Time course of chondroitin sulfate proteoglycan (CSPG) sulfation in the remote myocardium (non-scar tissue) after myocardial infarction (MI) caused by ischemia/reperfusion (I/R).

Superior cervical ganglion (SCG) explants from neonatal mice were co-cultured with left ventricle tissue. Although SCG contain a relatively small number of cardiac sympathetic neurons, they provide a well-characterized model for axon outgrowth studies (*Gardner and Habecker, 2013*). Ganglion explants co-cultured with unoperated left ventricle displayed uniform axon outgrowth in all directions while explants cultured alongside cardiac scar tissue (10–14 days post-I/R) exhibited shorter axons growing in the direction of the scar tissue (*Figure 3B–D*). Interestingly, ARSB treatment (0.6 µg/mL) of scar tissue fully restored axon outgrowth towards the scar (*Figure 3D*), while growth away from the scar tissue was normal in all conditions (*Figure 3E*). ARSB treatment of control explants had no effect on axon outgrowth, suggesting its effects are through altering the cardiac scar rather than the ganglion. Together, these data suggest that reducing 4S of CS-GAGs in the cardiac scar is sufficient to enable sympathetic axon regeneration.

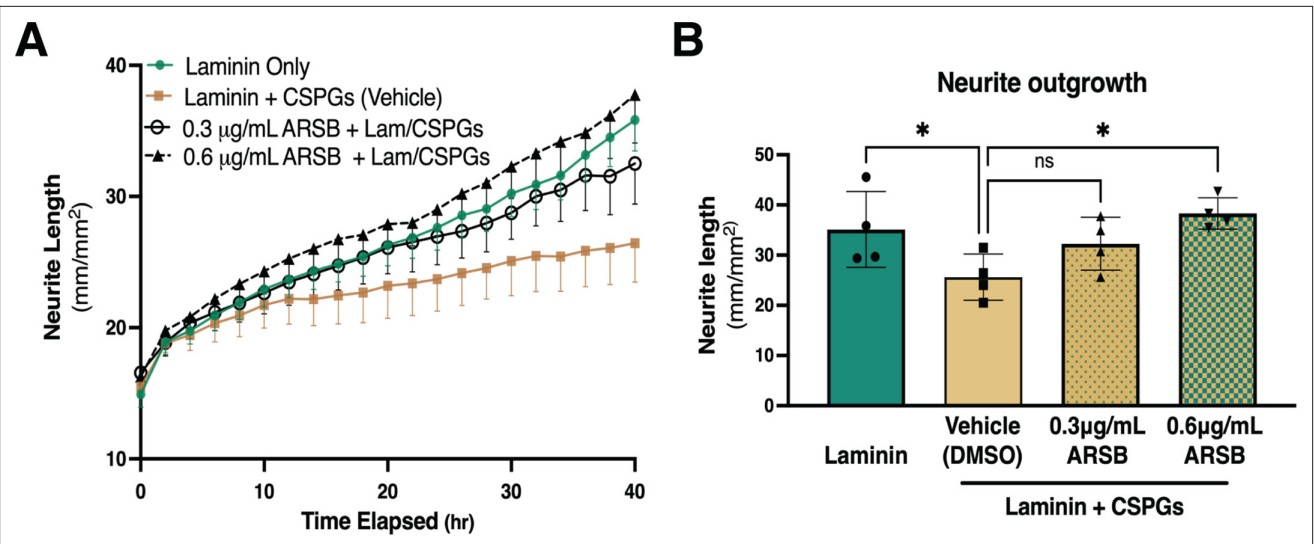

**Figure 2.** Reducing CS-GAG 4-sulfation promotes sympathetic neurite outgrowth in vitro. (**A**) Example of neurite outgrowth experiment with arylsulfatase-B (ARSB). Data are mean neurite length ± SD at 9 locations per well and 3 wells per condition. (**B**) Removal of 4-sulfation with ARSB restores neurite outgrowth to control levels. Quantification of dissociated sympathetic neurite length at 40 hr post-plating on indicated plate coatings. Data are mean neurite length ± SD; one-way ANOVA (Dunnett's post-test). All comparisons made to neurite outgrowth on laminin + CSPG; 0.3 μg/mL ARSB ns – not significant p-value = 0.193, laminin only *p=0.039, 0.6 μg/mL ARSB *p-value = 0.042, n=4 experiments.

The online version of this article includes the following figure supplement(s) for figure 2:

**Figure supplement 1.** Arylsulfatase-B (ARSB) removes 4S CS-GAGs and leaves 6S CS-GAGs intact.

## Expression of CSPG sulfation enzymes is altered after I/R

To understand the mechanisms by which CS-GAG sulfation increases after I/R in mice, we examined protein levels of three critical CSPG sulfation enzymes, comparing post-MI expression to unoperated left ventricle. We examined chondroitin-4-sulfotransferase CHST11, 4S-dependent chondroitin-6-sulfotransferase CHST15 (4,6-tandem sulfation enzyme), and 4-sulfatase ARSB (*Figure 1A*). CHST11 levels decreased significantly within 24 hr, persisting until day 7 (*Figure 4A and D*), while CHST15 was increased significantly at days 7 and 14 post-MI (*Figure 4B and D*). ARSB levels decreased significantly by day 3 and remained low through day 14 (*Figure 4C and D*).

In light of protein changes in CSPG sulfation enzymes after I/R, we examined CS-GAG sulfation in the mouse infarct at these same time points. 4S and 6S CS-GAGs are significantly increased in the infarct by day 7 post-MI extending through day 14 (*Figure 5A, B and E*). We asked if the CSPG core protein NG2 (also called CSPG4) was also altered after I/R, and found that NG2 was increased in the infarct by 7 days after MI (*Figure 5C and E*). Thus, core proteins and sulfation of CS-GAG side chains are both in greater abundance. We chose to look at NG2 because it was prominently expressed in a recent glycoproteomics analysis of cardiac scar tissue whereas another commonly studied core proteoglycan, Aggrecan, was not detected (*Tian et al., 2014*). In an effort to better understand how this process may be coordinated, we examined the expression of a key wound healing regulator, Galectin-3, which has also been connected to CSPG sulfation and ARSB expression (*Bhattacharyya et al., 2020*; *Bhattacharyya et al., 2017*; *Bhattacharyya et al., 2014*; *Bhattacharyya et al., 2015*). Galectin-3 was increased significantly by 24 hr after I/R, which may contribute to altered expression of CSPG sulfation enzymes at later time points (*Figure 5D and E*). These data suggest that increased CHST15 and decreased ARSB expression contribute to increased 4,6-sulfation of CS-GAGs observed by day 7 post-MI.

## siRNA knockdown of Chst15 reduces 4,6-tandem sulfation of CSPGs and restores sympathetic axons in the cardiac scar after I/R

To determine if CSPG sulfation suppresses sympathetic nerve regeneration after I/R in mice, we decreased 4,6-tandem sulfated CS-GAGs in vivo by reducing the expression of the *Chst15* gene using silencing RNA (siRNA). After identifying siRNA that decreased *Chst15* mRNA in myoblast-like C2C12

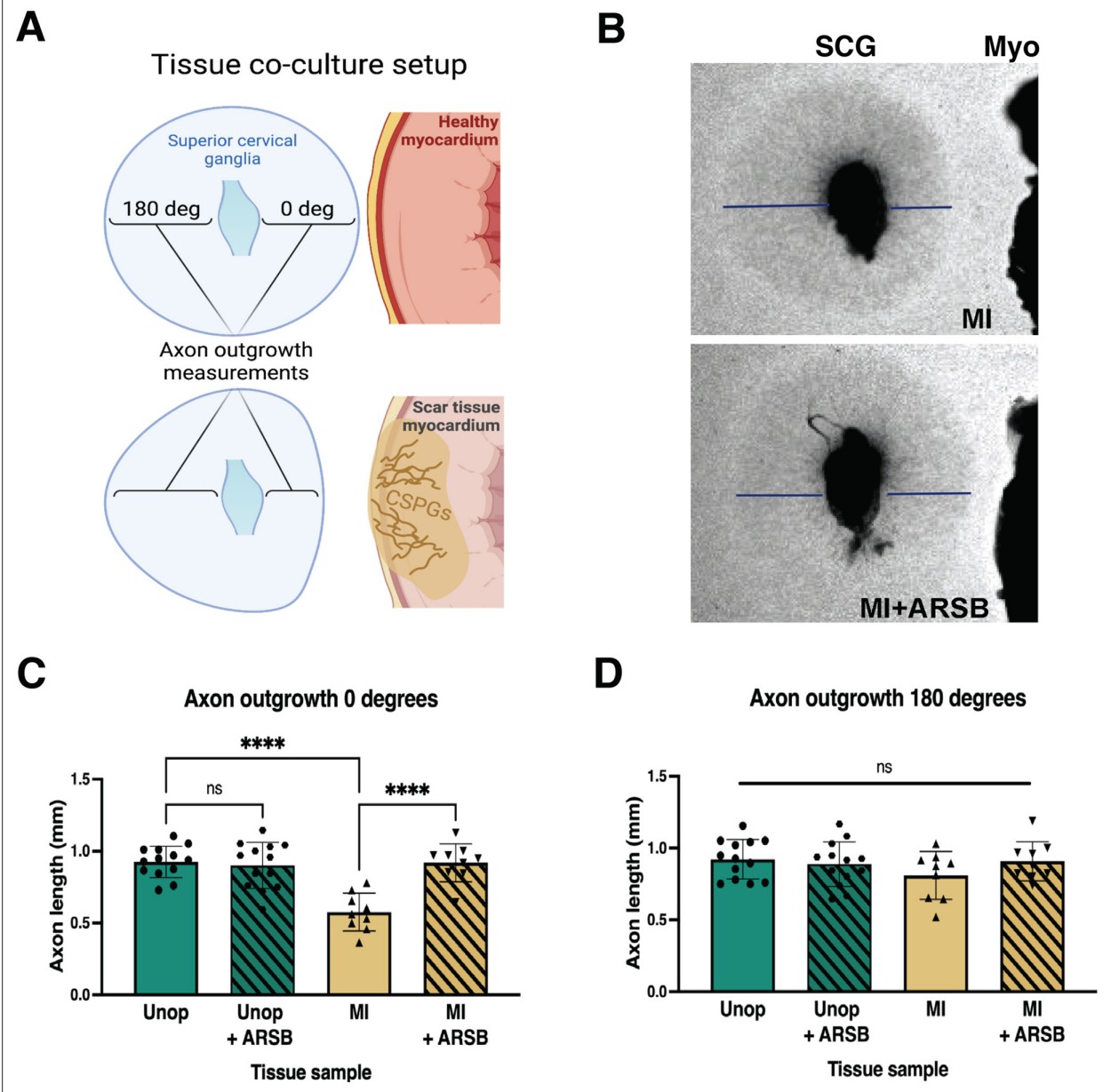

**Figure 3.** Reducing CS-GAG 4S in cardiac scar tissue ex vivo restores sympathetic axon outgrowth. (**A**) Explant co-culture schematic. Ganglion axon extension toward (0 degree) and away from (180 degree) myocardium was measured. (**B**) Example images of superior cervical ganglion (SCG) axon outgrowth in the presence of cardiac scar tissue (Myo) treated with or without arylsulfatase-B (ARSB). Lines show example measurement of axon extension 48 hr after plating. (**C**) Quantification of ganglion axon outgrowth toward myocardium of either healthy tissue (Unop) or scar tissue (myocardial infarction [MI]) treated with vehicle (5% DMSO) or ARSB (0.6 µg/mL). (**D**) Quantification of ganglion axon outgrowth away from myocardium of either healthy tissue (Unop) or scar tissue (MI) treated with vehicle (5% DMSO) or ARSB (0.6 µg/mL). Data are mean axon length ± SD. Statistics for C, D: one-way ANOVA (Dunnett's post-test), comparisons made to vehicle-treated MI tissue; 0 degree quantification ****$p < 0.0001$, 180 degrees quantification ns – not significant p-value = 0.204, 0.467, 0.355 left to right respectively, n=13 control tissue and n=9 MI tissue.

cells (*Figure 6—figure supplement 1A*,B), we tested the efficacy of our *Chst15* siRNA in vivo. Intravenous delivery of 100 µg si*Chst15* reduced CHST15 protein compared to non-targeting control siRNA (*Figure 6—figure supplement 1C*). Mice were then treated with 100 µg si*Chst15* on days 3, 5, and 7 post-MI, leading to a significant reduction in 4S,6S CS-GAGs. Specifically, si*Chst15* reduced the amount of 6S present 10 days post-MI compared to non-targeting siRNA control (*Figure 6A*). By 10 days after MI, CHST15 protein levels had returned to normal suggesting that transient knockdown of CHST15 is sufficient to alter 4,6-sulfation of CS-GAGs (*Figure 6—figure supplement 1D*,E).

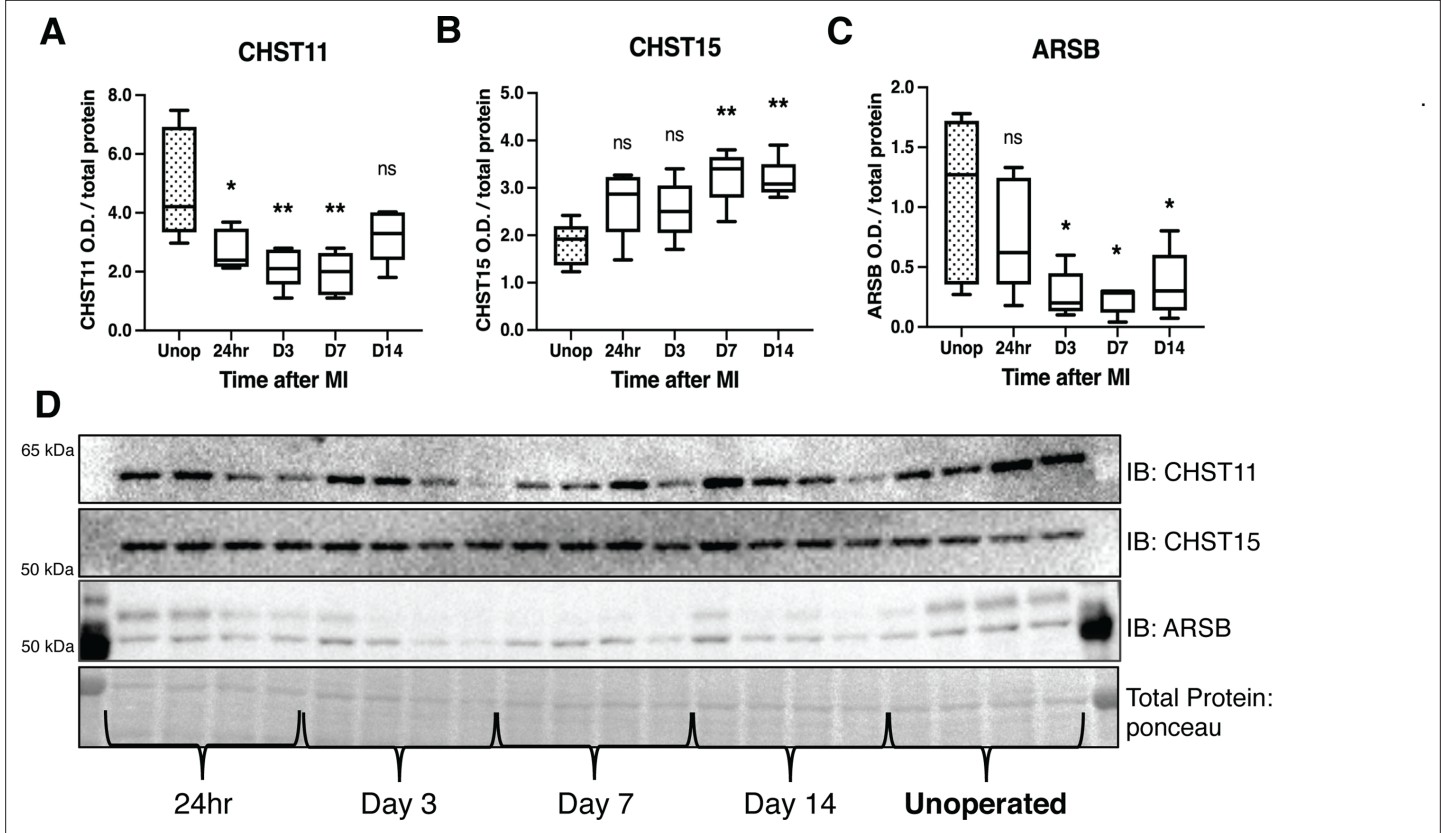

**Figure 4.** Chondroitin sulfate proteoglycans (CSPG) sulfation enzyme expression is altered after ischemia/reperfusion (I/R). Western blot quantification of protein expression for (**A**) chondroitin-4-sulfotransferase (CHST11), (**B**) chondroitin-6-sulfotransferase (CHST15), and (**C**) 4-sulfatase (arylsulfatase-B [ARSB]) protein expression in control left ventricle (Unop), or in the days after myocardial infarction (MI) (24 hr, D3, D7, or D14). Data are mean optical density (OD) ± SD. Statistics: one-way ANOVA (Dunnett's post-test) comparisons to unoperated tissue; CHST11 24 hr *p-value = 0.015, D3 **p-value = 0.002, D7 **p-value = 0.001, D14 ns – not significant p-value = 0.068; CHST15 24 hr ns p-value = 0.073, D3 ns p-value = 0.164, D7 **p-value = 0.002, D14 **p-value = 0.004; ARSB 24 hr ns p-value = 0.554, D3 *p-value = 0.018, D7 *p-value = 0.012, D14 *p-value = 0.035; n=5 animals per group. (**D**) Example western blot images for CSPG sulfation enzymes.

Interestingly, si*Chst15* knockdown led to increased NG2 core protein expression compared to non-targeting siRNA control (**Figure 6C**). The sympathetic neuron marker Tyrosine Hydroxylase (TH) was increased significantly in cardiac scar tissue from si*Chst15*-treated animals compared to non-targeting siRNA control animals (**Figure 6D**), suggesting successful sympathetic reinnervation of the cardiac scar.

To ensure that reinnervation of the cardiac scar occurred after *Chst15* knockdown, we examined the infarct (labeled by fibrinogen) for TH-positive sympathetic nerve fibers by immunohistochemistry (IHC). IHC analysis in animals treated with a non-targeting siRNA control showed clear denervation of the infarct (**Figure 7A**) compared to a peri-infarct region adjacent to the scar (**Figure 7B**). Animals treated with an siRNA targeting *Chst15* had restored TH-positive fibers in the infarct (**Figure 7C**) at the same innervation density as the peri-infarct region (**Figure 7D**). TH innervation density was significantly higher in the infarct of *Chst15*-targeted animals compared to non-targeting siRNA-treated animals (**Figure 7E**). Cardiac scar size was examined between the two groups and no significant difference existed (**Figure 7G and H**). To examine whether restoration of nerves reduced arrhythmia susceptibility, we examined arrhythmias by ECG after administration of the β-agonist isoproterenol and caffeine (**Gardner et al., 2015**; **Wang et al., 2014**). Restoration of nerves with *Chst15* siRNA treatment reduced arrhythmias when compared to non-targeting siRNA control-treated animals (**Figure 7F**). These results indicate that reducing 4,6-tandem sulfated CS-GAGs in the cardiac scar after MI promotes sympathetic nerve regeneration back into the cardiac scar and reduces arrhythmia susceptibility after MI.

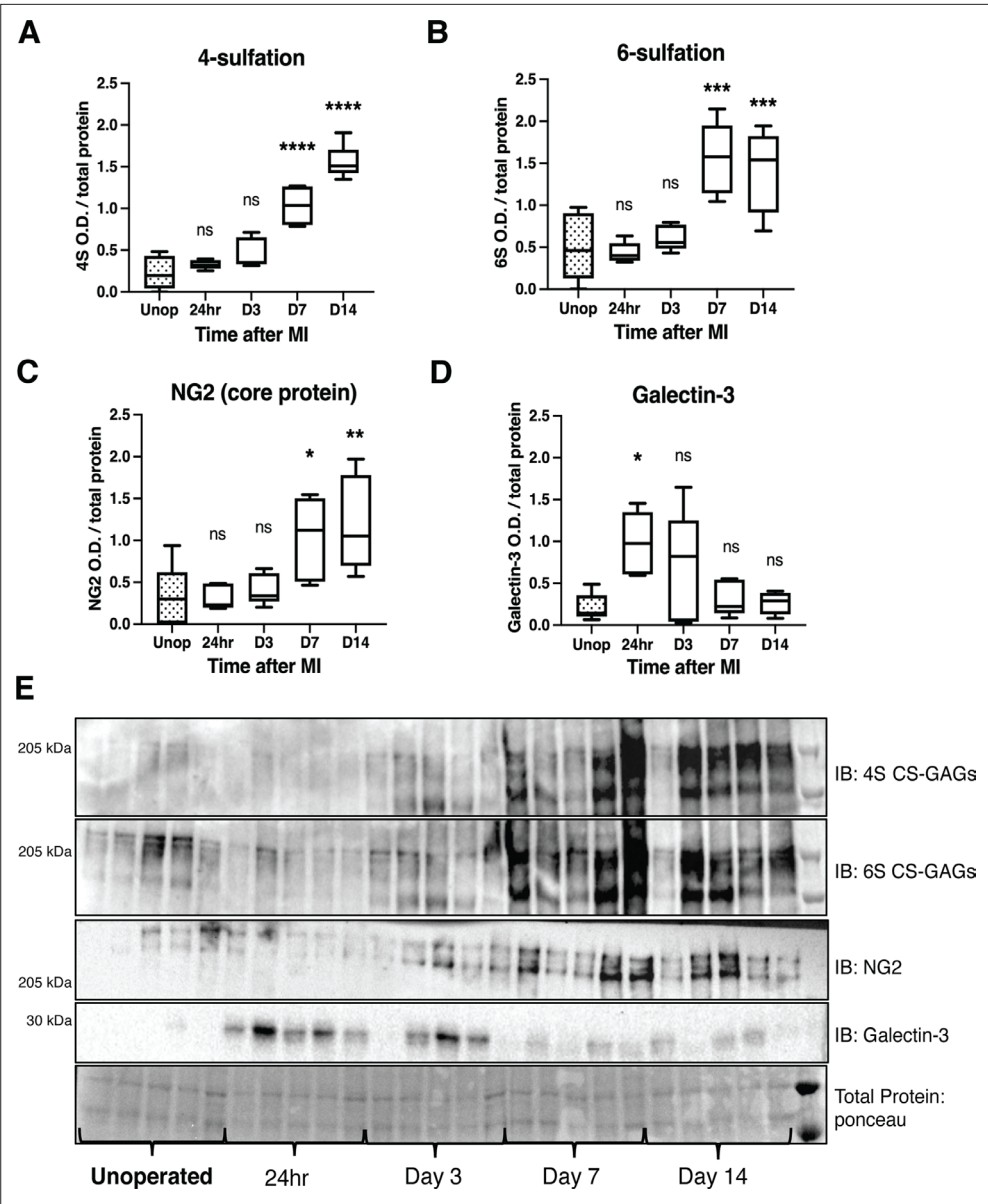

**Figure 5.** Time course of chondroitin sulfate proteoglycan (CSPG) sulfation and core protein expression after ischemia/reperfusion (I/R) in cardiac scar tissue. Western blot quantification of (**A**) 4-sulfation (4S CS-GAGs), (**B**) 6-sulfation (6S CS-GAGs), (**C**) NG2 core protein, and (**D**) Galectin-3 in the days after MI. Data are mean optical density (OD) ± SD. Statistics: one-way ANOVA (Dunnett's post-test), comparisons to unoperated tissue; 4S, 24 hr ns – not significant p-value = 0.804, D3 ns p-value = 0.124, D7 ****p-value < 0.0001, D14 ****p-value < 0.0001; 6S, 24 hr ns p-value = 0.999, D3 ns p-value = 0.905, D7 ***p-value = 0.0001, D14 ***p-value = 0.0003; NG2, 24 hr ns p-value = 0.999, D3 ns p-value = 0.977, D7 *p-value = 0.031, D14 **p-value = 0.007; Galectin-3, 24 hr *p-value = 0.013, D3 ns p-value = 0.172, D7 ns p-value = 0.974, D14 ns p-value = 0.998; n=5 animals per group. (**E**) Example western blot images of A–D.

## Discussion

This study investigated the role of CS-GAG side chain sulfation in preventing sympathetic axon regeneration in the heart after MI. We found that the CSPGs present in the cardiac scar after ischemia-reperfusion were enriched with sulfation at the 4 and 6 positions of CS-GAGs, which is often referred to as chondroitin sulfate E (CS-E) (*Miller and Hsieh-Wilson, 2015*). CS-E inhibits the outgrowth of several types of neurons in the CNS (*Brown et al., 2012*; *Gilbert et al., 2005*), and our data indicate that it also prevents growth of sympathetic axons, since decreasing either 4S or 6S allowed axon

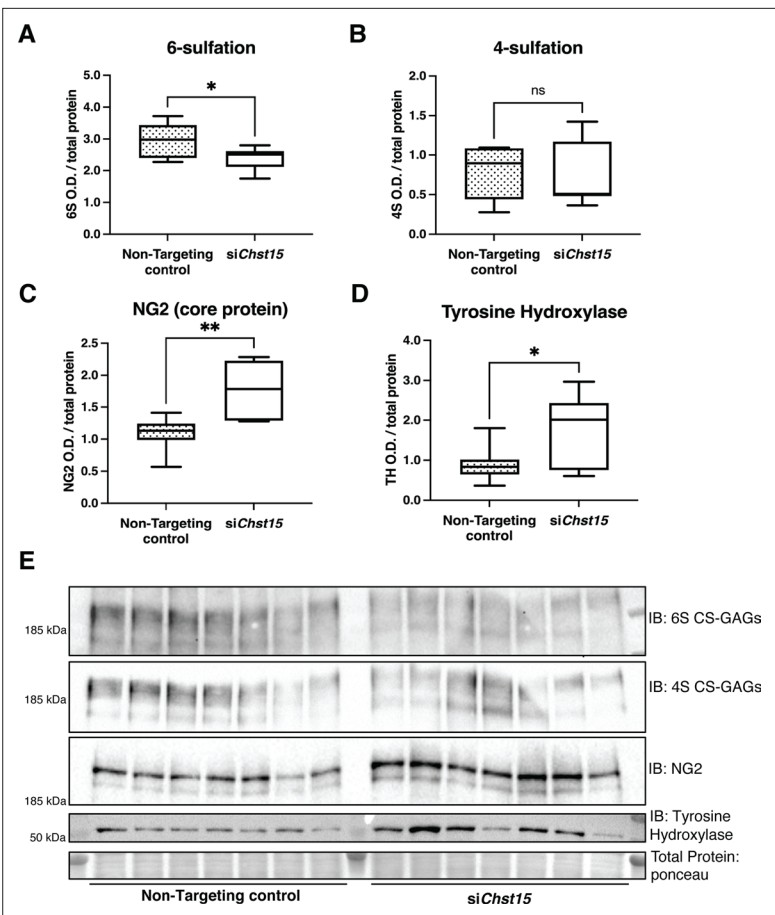

**Figure 6.** Reducing CS-GAG 4,6-tandem sulfation in vivo promotes sympathetic nerve regeneration into the cardiac scar after ischemia/reperfusion (I/R). Western blot quantification of (**A**) 6-sulfation (6S CS-GAGs), (**B**) 4-sulfation (4S CS-GAGs), (**C**) NG2 core protein, and (**D**) the sympathetic neuron marker Tyrosine Hydroxylase (TH) in the cardiac scar after transient *Chst15* knockdown or treatment with a non-targeting silencing RNA (siRNA) control. Tissue was collected 10 days after myocardial infarction (MI), n=7 animals per group. Data are mean optical density (OD) ± SD. Statistics: Student's t-test (Welch's test); 6S, *p-value = 0.028; 4S, ns – not significant p-value = 0.826; NG2, **p-value = 0.004; TH, *p-value = 0.043. (**E**) Western blot images of A–D.

The online version of this article includes the following figure supplement(s) for figure 6:

**Figure supplement 1.** Identification of an effective silencing RNA (siRNA) against *Chst15*.

regeneration. Nerve regeneration was confirmed by histology and by expression of the neuronal protein TH in cardiac scar tissue. Reinnervation throughout the left ventricle decreased isoproterenol-induced arrhythmias, consistent with previous studies targeting the neuronal CSPG receptor PTPσ (*Gardner et al., 2015*; *Sepe et al., 2022*). We do not fully understand the mechanisms by which reinnervation decreases arrhythmia susceptibility, but reinnervation of borderzone myocytes may play an important role. In addition, therapeutics targeting PTPσ to restore innervation shift the inflammatory response toward a more reparative phenotype, and in some cases decrease infarct size (*Sepe et al., 2022*). These new data show that targeting CSPGs in the scar to restore innervation decreases arrhythmia susceptibility without altering infarct size. It remains unknown if modulating CSPG sulfation alters the inflammatory response.

The core protein NG2 is an important source of CSPGs in the cardiac scar (*Tian et al., 2014*), and there is evidence that the NG2 core protein can inhibit sensory neurite extension independent of CS sulfation in certain contexts (*Dou and Levine, 1994*; *Ughrin et al., 2003*). Modulating the cardiac scar is of interest as a strategy to improve outcomes after MI, but removing matrix proteins presents a challenge since the scar plays an important role in maintaining structural integrity. Given the critical role of CSPG sulfation in other contexts (*Brown et al., 2012*; *Gilbert et al., 2005*; *Miller and Hsieh-Wilson,*

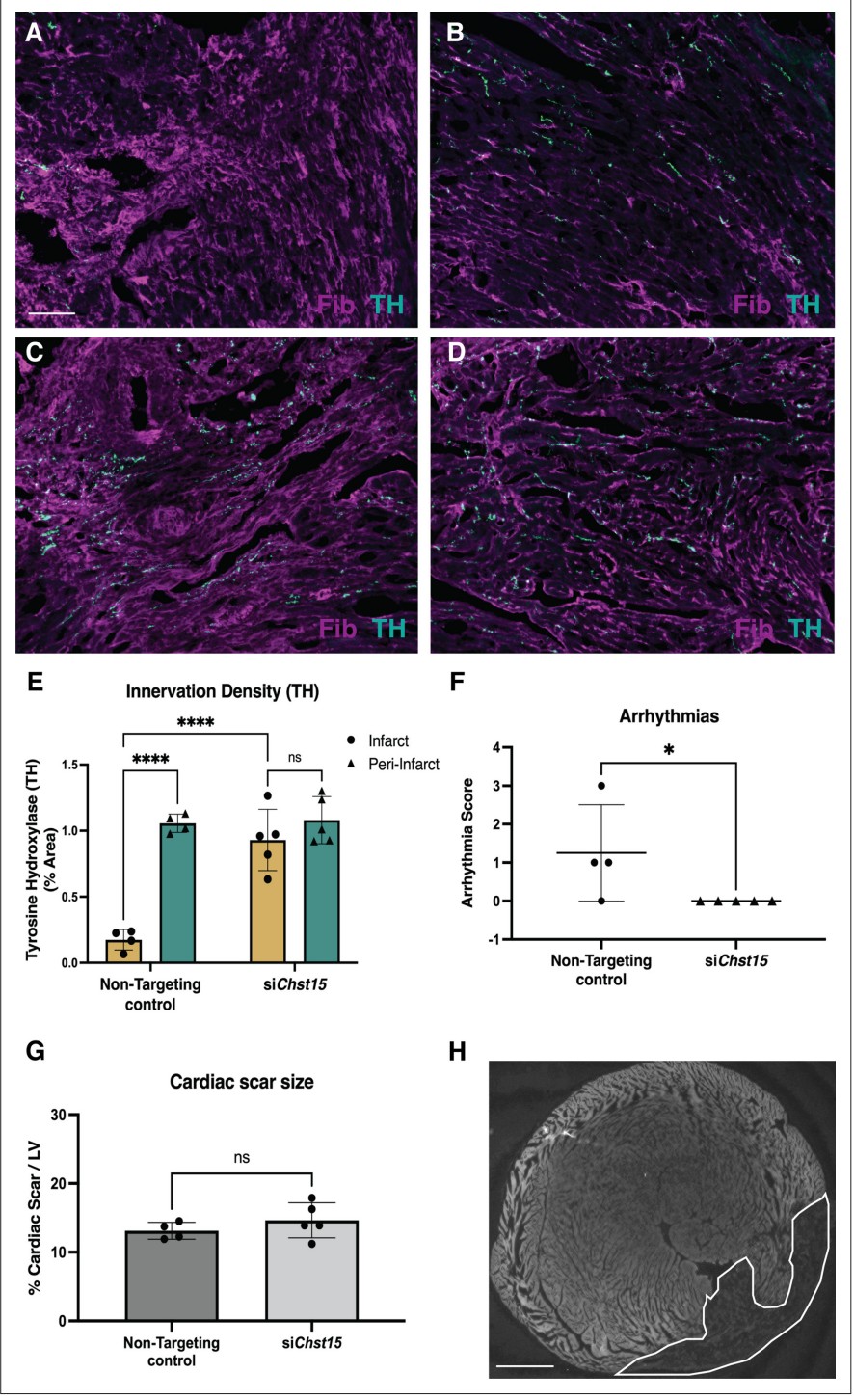

**Figure 7.** Reducing 4,6-tandem sulfation of CS-GAGs in vivo promotes sympathetic nerve regeneration into the cardiac scar. Fibrinogen was used to label scar (magenta) and Tyrosine Hydroxylase (TH) was used to label sympathetic neurons (cyan). Example image of (**A**) denervated infarct in non-targeting control-treated animal versus (**B**) normal density of TH + fibers in the peri-infarct region adjacent to the scar. Systemic delivery of silencing RNA (siRNA) against *Chst15* post-MI restored TH + fibers in the cardiac infarct (**C**) without altering nerve density outside the infarct (**D**). Scale bar, 100 μm. (**E**) Quantification of TH innervation density 10 days after MI. n=4 animals for non-targeting controls and n=5 for *Chst15* siRNA-treated animals. Data are mean percent TH ± SD. Two-way ANOVA, Tukey's post-test to compare all groups, select comparisons shown; si*Chst15* infarct vs. peri-nfarct ns – not significant p-value = 0.668, non-targeting infarct vs. peri-nfarct ****p-value < 0.0001, non-targeting infarct

*Figure 7 continued on next page*

*Figure 7 continued*

vs. si*Chst15* infarct ****p-value < 0.0001. (**F**) Arrhythmia scores based on the most severe arrhythmia observed in each heart after injection of isoproterenol and caffeine (0=no PVCs, 1=single PVCs, 2=bigeminy or salvos, 3=non-sustained ventricular tachycardia). See Materials and methods for details. Treatment with siRNA against *Chst15* reduced arrhythmias compared to non-targeting controls. n=4 animals for non-targeting controls and n=5 animals for *Chst15 siRNA*. Data are arrhythmia score for each animal. Statistics: Student's t-test (Mann-Whitney test), *p<0.047. (**G**) Cardiac scar size assessed as a percent area of total left ventricle (LV) area was unaltered in *Chst15* siRNA-treated animals compared to non-targeting controls. Quantification of n=4 animals for non-targeting controls and n=5 for *Chst15 siRNA*-treated animals. Data are mean percent area scar ± SD. Statistics: Student's t-test (Welch's test), ns – not significant p-value = 0.283. (**H**) Example 2× image of cardiac scar in *Chst15* siRNA-treated heart using autofluorescence, absence of autofluorescence indicates region of the infarct, outlined in white. Scale bar, 500 µm.

---

*2015*; *Pearson et al., 2018*; *Wang et al., 2008*; *Yi et al., 2012*), we postulated that this might present an intervention point. While our data indicate that NG2 expression increases following MI, we found that removing or preventing sulfation of CS-GAG chains was sufficient to allow axon regeneration into the cardiac scar. Thus, NG2 prevents sympathetic axon regeneration in the heart exclusively through 4,6-tandem sulfation of CS side chains.

Increased 4,6-sulfation of CS-GAGs coincided with increased expression of the 4S-dependent 6-sulfotransferase CHST15, which is the final enzyme in the production of CS-E, and depletion of the sulfatase ARSB which removes 4S. These changes in enzyme expression provide a potential mechanism for the shift to highly sulfated GAG side chains after MI. Regulation of these proteins is also linked in the context of TBI and other diseases with a downregulation of ARSB leading to increased CHST15 (*Bhattacharyya et al., 2020* ; *Bhattacharyya et al., 2017*; *Bhattacharyya et al., 2014*; *Bhattacharyya et al., 2015*). Hypoxia reduces ARSB activity, increasing the abundance of 4-sulfated CS-GAGs which enables Galectin-3 mediation of transcriptional changes (*Bhattacharyya et al., 2014*). Therefore, we asked if Galectin-3 expression was altered in the heart after ischemia-reperfusion. To our surprise, we found that Galectin-3 increased before changes were observed expression of sulfation-related enzymes. This suggests that increased Galectin-3 may be an early driver of the accumulating highly sulfated CS-GAGs observed by day 7 post-MI.

The inhibitory effects of CS-E on axon outgrowth in certain neuronal subtypes led many to hypothesize that reducing CHST15 activity would restore nerve growth across CSPGs (*Brown et al., 2012*). To that end at least one group has even generated a novel small molecule to inhibit CHST15 and promote nerve growth (*Cheung et al., 2017*). However, since no commercially available inhibitor exists to our knowledge, we sought a different method. Targeting *Chst15* gene expression with siRNA decreased 4,6-tandem sulfation of CS-GAGs and allowed nerve regeneration into the cardiac scar. We were inspired to test *chst15* siRNA by a colitis study in mouse showing that siRNA knockdown of *chst15* reduced CS-E (*Suzuki et al., 2016*; *Suzuki et al., 2017*). More importantly, *Chst15* siRNA has been tested in a Phase 1 Clinical trial for Crohn's disease patients with active mucosal lesions, and a Phase I/IIa trial for patients with pancreatic cancer (*Tsuchiya et al., 2021*), and has a good safety profile (*Suzuki et al., 2017*). Thus, siRNA knockdown of *Chst15* is potentially translatable to large animals and eventually human studies. We treated mice on days 3, 5, and 7 post-MI with siRNA as we wanted to intervene while the cardiac scar was forming. We assessed nerve regeneration into the scar and CSPG sulfation on day 10, which is 3 days after the final injection of siRNA. Three days after the last siRNA injection, CHST15 levels were similar in non-targeting siRNA and *Chst15* siRNA hearts despite a significant reduction in 6S and reinnervation of the infarct in *Chst15* siRNA-treated animals. These data indicate that a transient decrease in CHST15 protein and depletion of CS-GAG sulfation is sufficient to allow nerve restoration in the cardiac scar where NGF is abundant. A limitation of this method is that it cannot rule out a systemic effect of *Chst1*5 siRNA, so we also attempted to use an adeno-associated virus 9 (AAV9) virus to elevate ARSB expression specifically in the heart after MI. However, we were unable to increase ARSB expression in the heart, although the AAV9-GFP (green fluorescent protein) control generated good expression in myocardium. ARSB is an FDA-approved therapeutic (Naglazyme) to treat mucopolysaccharidosis IV disease (*Harmatz et al., 2005*; *Harmatz et al., 2004*; *Muñoz-Rojas et al., 2010*), which may be of use in the therapeutic context of the post-MI heart. Ultimately this work demonstrates the critical nature of 4,6-sulfation of CS-GAGs in preventing

reinnervation of the heart after MI and provides two strategies to promote reinnervation by modulating CSPG sulfation.

# Materials and methods

## Key resources table

| Reagent type (species) or resource | Designation | Source or reference | Identifiers | Additional information |
|---|---|---|---|---|
| Strain, strain background (*Mus musculus*) | C57BL/6J | Jackson Laboratories | 000664 | Male and female mice used for I/R surgeries |
| Strain, strain background (*Rattus norvegicus*) | Crl:CD(SD) Outbred | Charles River Laboratories | 001 | Newborn pups, P0-P2, male and female for sympathetic neuron cultures |
| Cell line (*Mus musculus*) | C2C12 myoblast | ATCC | CRL-1772 | siRNA knockdown pilot studies |
| Antibody | Anti-CHST11 (Rabbit polyclonal) | Invitrogen | PA5-68129 | WB (1:500) |
| Antibody | Anti-CHST15 (Rabbit polyclonal) | Proteintech | 14298-1-AP | WB (1:1000) |
| Antibody | Anti-ARSB (Rabbit polyclonal) | Proteintech | 13227-1-AP | WB (1:500) |
| Antibody | Anti-NG2 (Rabbit polyclonal) | Millipore | AB5320 | WB (1:1000) |
| Antibody | Anti-Tyrosine hydroxylase (Rabbit polyclonal) | Millipore | AB152 | WB (1:1000) IF (1:1000) |
| Antibody | Anti-Galectin-3 (Mouse monoclonal) | Abcam | AB2785 | WB (1:1000) |
| Antibody | Anti-chondroitin-4-sulfate (Mouse monoclonal) | Millipore | MAB2030 | WB (1:1000) |
| Antibody | Anti-chondroitin-6-sulfate (Mouse monoclonal) | Millipore | MAB2035 | WB (1:1000) |
| Antibody | Anti-Fibrinogen (Sheep polyclonal) | BioRad | 4400-8004 | IF (1:300) |
| Antibody | Anti-Rabbit IgG secondary (Goat polyclonal) | Molecular Probes | A-11034 | IF (1:1000) |
| Antibody | Anti-Sheep IgG secondary (Donkey polyclonal) | Molecular Probes | A-21099 | IF (1:1000) |
| Antibody | Anti-Rabbit HRP-conjugated secondary (Mouse polyclonal) | Thermo Fisher Scientific | A16104 | WB (1:10,000) |
| Antibody | Anti-Mouse HRP-conjugated secondary (Goat polyclonal) | Thermo Fisher Scientific | 31430 | WB (1:10,000) |
| Other | Ponceau S | Thermo Fisher Scientific | A40000278 | 3–5 min stain, de-stain with ddH$_2$O |
| Peptide/ recombinant protein | PageRuler Plus prestained ladder | Thermo Fisher Scientific | 26619 | (7 μL) per well |
| Other | Poly-L-Lysine | Sigma-Aldrich | P8920 | (0.01%) for plate coating |
| Other | Laminin | RND Systems | 3400-010-02 | (1 μg/mL) |
| Other | Soluble CSPGs | Millipore | CC117 | (2 μg/mL) |
| Peptide/recombinant protein | NGF | Alomone Labs | N-100 | (10 ng/mL) |
| Peptide/recombinant protein | Recombinant ARSB | RND Systems | 4415-SU-010 | (0.6 μg/mL) |
| Peptide/recombinant protein | Recombinant Chondroitinase ABC | RND Systems | 6877 GH-020 | (100 μU/mL) |
| Chemical compound, drug | Dharmafect | Horizon Discovery | T-2001-02 | Use manufacturer's protocol |

*Continued on next page*

*Continued*

| Reagent type (species) or resource | Designation | Source or reference | Identifiers | Additional information |
|---|---|---|---|---|
| Sequence-based reagent | siRNA: *Chst15* | Horizon Discovery | J-059417-09 | (120 nM) – in vitro<br>(100 µg) – in vivo<br>*Accell* formulation |
| Sequence-based reagent | siRNA: Non-targeting | Horizon Discovery | D-001910–01 | (120 nM) – in vitro<br>(100 µg) – in vivo<br>*Accell* formulation |
| Sequence-based reagent | *Chst15* primer | Thermo Fisher Scientific | 4331182 | Manufacturer's protocol |
| Sequence-based reagent | *Gapdh* primer | Thermo Fisher Scientific | 4448489 | Manufacturer's protocol |
| Other | C18 Column | Agilent | AG-588945-902 | (50 × 4.6 mm, 5 µm) |
| Chemical compound, drug | Isoproterenol | Sigma-Aldrich | CAS:5985-95-2 | (50 µg) |
| Chemical compound, drug | Caffeine | Sigma-Aldrich | CAS:58-08-2 | (3 mg) |
| Software, algorithm | Prism | GraphPad | Version 9 | Statistical analysis |
| Software, algorithm | ImageJ | NIH | https://imagej.net/ | Image analysis |
| Other | BZ-X 800 | Keyence | | Imaging of co-culture (2×) and cardiac tissue sections (20×) |
| Other | Incucyte | Essen Biosciences | | Imaging neurite outgrowth (20×) |

## Animals

C57BL/6J mice obtained from Jackson Laboratories West (Sacramento, CA) were used for all experiments. All mice were kept on a 12 hr:12 hr light-dark cycle with ad libitum access to food and water. Age and gender-matched male and female mice 12–18 weeks of age were used for myocardial-ischemia reperfusion surgeries, as described below. Pregnant Sprague-Dawley rats were purchased from Charles River Laboratories (Wilmington, MA) for experiments with dissociated sympathetic neurons. SCG from male and female neonatal rats (P0-P2) were used to generate primary sympathetic neuron cultures. All rats were kept on a 12 hr:12 hr light-dark cycle with ad libitum access to food and water. All procedures were approved by the OHSU Institutional Animal Care and Use Committee and comply with the Guide for the Care and Use of Laboratory Animals published by the National Academies Press (8th edition).

## Myocardial I/R surgery

Anesthesia was induced with 4% isoflurane and maintained with 2% isoflurane. Mice were restrained supine, intubated, and mechanically ventilated. Core body temperature was monitored by a rectal probe and maintained at 37°C throughout the surgery. The left anterior descending coronary artery (LAD) was reversibly ligated for 40 min and then reperfused by release of the ligature. LAD occlusion was verified by a persistent S-T wave elevation, region-specific cyanosis, and wall motion abnormalities. Reperfusion was confirmed by return of S-T wave to baseline level and re-coloration of ventricle region distal to occlusion (*Gardner and Habecker, 2013*; *Parrish et al., 2010*).

## Western blotting

Cardiac scar tissue from the left ventricle was dissected at 24 hr, 3 days, 7 days, and 14 days following MI, and control left ventricle tissue from unoperated animals. Heart tissue was pulverized in a glass douncer in NP40 lysis buffer (50 mM Tris [pH 8.0], 150 mM NaCl, 2 mM EDTA, 10 mM NaF, 10% glycerol, and 1% NP-40) containing complete protease inhibitor cocktail (Roche), phosphatase inhibitor cocktails 2 and 3 (Sigma). Lysates sat on ice for 30 min with intermittent vortexing. Lysates were centrifuged (13 k rpm, 10 min, 4°C) and resolved on 4–12% Bis-Tris gradient gel (3–8% Tris-Acetate gel for *Figure 5* sulfation studies) by SDS/PAGE, transferred to nitrocellulose membrane (GE Life Sciences), blocked in 5% nonfat milk, probed with either CHST11 (1:500; Invitrogen: PA5-68129), CHST15 (1:1000; Proteintech: 14298-1-AP), ARSB (1:500; Proteintech: 13227-1AP), NG2/CSPG4 (1:1000; Millipore: AB5320), TH (1:1000; Millipore: AB1542), or Galectin-3 (1:500; Abcam: AB2785) then probed with

goat anti-rabbit (or mouse) HRP-conjugated secondary antibody (1:10,000; Thermo), and detected by chemiluminescence (Thermo Scientific). To detect sulfation of chondroitin-sulfate proteoglycans, lysates were treated according to the detailed deglycosylation protocol from Mariano Viapiano PhD lab website (*Massey et al., 2008*). Briefly, 50–100 µg of protein lysate was treated for 6–8 hr with chABC (100 µU/mL; R&D systems) before being subjected to traditional western blotting, probed with anti-chondroitin-4-sulfate (1:1000; Millipore: MAB2030) or anti-chondroitin-6-sulfate (1:1000; Millipore: MAB2035). Protein expression was quantified with ImageJ densitometry and normalized to total protein as measured by Ponceau staining.

## Sympathetic outgrowth assay

Cultures of dissociated sympathetic neurons were prepared from SCG of newborn rats as described (*Dziennis and Habecker, 2003*). Cells were pre-plated for 1 hr to remove non-neuronal cells, and then 5000 neurons/well were plated onto a 96-well plate (TPP) coated with (poly-L-lysine, 0.01%, Sigma-Aldrich) and either laminin (1 µg/mL, RND systems), laminin and CSPGs (2 µg/mL; Millipore), or laminin and CSPGs pre-treated with ARSB (0.3 or 0.6 µg/mL; R&D systems) for 6–8 hr. Neurons were cultured in serum free C2 medium (*Pellegrino et al., 2011*) supplemented with 10 ng/mL NGF (Alomone Labs), 100 U/mL penicillin G, and 100 µg/mL streptomycin sulfate (Invitrogen). Live cell imaging was carried out using an Incucyte Zoom microscope (Essen BioScience), with 20× phase images acquired every 2 hr over a 40 hr period. Neurite length was measured using Cell Player Neurotrack software (Essen BioScience) and was used to calculate the neurite growth rate.

## Explant co-culture assay

Explants were generated as previously described (*Gardner and Habecker, 2013*). Briefly, SCG were dissected from neonatal mice and placed into pre-marked tissue culture wells so that ganglia were approximately 1 mm from left ventricle tissue. LV tissue was taken from unoperated control animals or from hearts collected 10–14 days after I/R. Cardiac tissue from a single animal was split in half and cultured with the left or right ganglion from a single animal, this enabled tissue from the same mice to be treated with either vehicle (5% DMSO) or ARSB to remove 4S-CS GAGs. The ganglia and the cardiac tissue were co-cultured inside a Matrigel bubble surrounded by C2 media supplemented with 2 ng/mL NGF to ensure neuron survival and stimulate a basal level of axon outgrowth in all directions. ARSB was added directly to the media (0.6 µg/mL) at the time of plating and again 24 hr later. At 48 hr the explants were imaged by phase microscopy with a Keyence BZ-X microscope. Neurite length from the edge of the ganglia to the most distal tip of visible neurites was measured using ImageJ.

## siRNA pool knockdown efficiency screen

A pool of four siRNAs targeting the *Chst15* gene were purchased from Horizon Discovery (formerly GE Dharmacon) and were tested for their efficacy in reducing gene expression in C2C12 myoblasts. C2C12 myoblasts were plated on 12-well plates coated with collagen and were transfected with the various targeting and control siRNAs using the Dharmafect transfection reagent (3 µL of Dharmafect reagent per well with 120 nM siRNA). The next day wells were split using Versene (Gibco), one-third of the cells were collected for a 24 hr time point and the other two-thirds were split into two wells for 48 and 72 hr knockdown time points. Cells were processed using an RNA Mini-kit (Qiagen) to purify RNA. One µg of RNA was loaded into the cDNA reaction with the iScript cDNA synthesis kit. Gene expression was examined with multiplexed Taqman probes targeting *Chst15* and *Gapdh* using 2 µL of cDNA template and Taqman reagents and measured with an ABI7500 Thermocycler. The delta delta ct method was used to calculate knockdown efficiency. Once an effective siRNA against *Chst15* was identified, knockdown efficiency was measured at the protein level with C2C12 myoblasts 48 hr post-knockdown. CHST15 expression was assessed via western blot with a CHST15 antibody (1:1000; Proteintech: 14298-1-AP) normalized to GAPDH (1:1000; Thermo Scientific: MA1-16757); siRNA *Chst15-2* was selected for larger scale production and used for in vivo studies.

## In vivo siRNA treatment

After MI surgery mice were treated with 100 µg of siRNA targeting *Chst15* or a non-targeting control. siRNA was delivered systemically via tail-vein injection on days 3, 5, and 7 following MI. Tissue was collected at day 10 post-MI for western blot analysis of CSPG sulfation and the sympathetic neuron

marker TH. siRNA used for this experiment was custom synthesized by Horizon Discovery; siAccell in vivo formulation.

## Norepinephrine content analysis by HPLC

Norepinephrine (NE) levels in heart tissue were measured by high-performance liquid chromatography (HPLC) with electrochemical detectionLi, Knowlton et al. 2004. Frozen, pulverized tissue was weighed and homogenized in 300 µL of 0.1 M perchloric acid (PCA) containing 0.5 µM dihydroxybenzylamine (internal standard). The homogenate was centrifuged at 14,000 rpm for 4 min and NE in 100 µL of the supernatant was adsorbed onto 15 mg alumina, followed by 15 min of tumbling. The alumina was washed twice with ddH$_2$O and the catechols desorbed with 150 µL of 0.1 M PCA. The catechols were separated by reversed-phase HPLC on a C18 column (Agilent Microsorb, 150 × 4.6 mm, 5 µm) and measured by an electrochemical detector (Coulochem III; ESA, Bedford, MA) with the electrode potential set at +180 mV as described previously (*Parrish et al., 2010*). The mobile phase used consisted of 75 mM sodium phosphate (pH 3.0), 1.7 mM sodium octane sulfonate, 3.0% acetonitrile. NE standards (0.5 µM) were processed in parallel with the tissue samples.

## Immunohistochemistry

TH (sympathetic nerve fibers) and fibrinogen (Fib; infarct/scar) staining was carried out as described previously (*Gardner et al., 2015*). Tissue was collected 10 days after surgery, fixed in 4% paraformaldehyde, frozen and 12 µm sections generated. To reduce autofluorescence sections were rinse 3 × 10 min in 10 mg/mL sodium borohydride and rinsed for 3 × 10 min in PBS. Slides were placed in 2% BSA, 0.3% Triton X-100 in PBS for 1 hr, and then incubated with rabbit anti-TH (1:1000; Millipore: AB1542) and sheep anti-fibrinogen (1:300; BioRad: 4400-8004) overnight. The following day the slides were incubated with Alexa-Fluor IgG-specific antibodies (Molecular Probes, 1:1000) for 1.5 hr and rinsed 3 × 10 min in PBS. Background autofluorescence was reduced further with a 30 min incubation in 10 mM CuSO$_4$ (diluted in 50 mM ammonium acetate). Following this, slides were rinsed 3 × 10 min in PBS before mounting in 1:1 glycerol:PBS and visualized by fluorescence microscopy. Threshold image analysis of TH staining has been described previously (*Gardner et al., 2015*) but briefly, the threshold function in ImageJ was used to generate black and white images discriminating TH + nerves for six sections spanning 200 µm of the infarct/scar or peri-infarct region from each heart. Percent area TH + fiber density (20× field of view) was quantified within the infarct and the area immediately adjacent to the infarct (peri-infarct). Images were acquired with a Keyence BZ-X 800 microscope.

## Quantification of infarct size

Infarct size was determined 10 days after myocardial I/R injury and tissue was prepared for IHC as previously mentioned, but omitting the steps to reduce autofluorescence. Autofluorescence was used to image the infarct size. Scar tissue is notably lacking autofluorescence enabling easy identification of the infarct. Images were acquired with a Keyence BZ-X 800 microscope at 2× magnification and analyzed using Image J freehand selection tool. Left ventricle (LV) and infarct was outlined, measured, and the percent area of cardiac scar was determined by (infarct area/LV area) × 100. The scar was imaged in six sections across 200 µm of the infarct as previously described (*Gardner et al., 2015*).

## Arrhythmia assessment

Anesthesia was induced with 4% isoflurane and maintained with 2% isoflurane in day 10 post-MI animals treated with siRNA. ECG leads were connected to monitor arrhythmias and animals were maintained at 37°C throughout the analysis. All parameters were monitored with Powerlab LabChart software (AD Instruments). A 30 min baseline was used to assess spontaneous arrhythmia susceptibility prior to administration of β-agonist isoproterenol (50 µg) and caffeine (3 mg) as described previously (*Wang et al., 2014*). Arrhythmias were measured for 30 min following drug administration and scored according to the modified Lambeth conventions (*Curtis et al., 2013*) on a scale of 0–4. Individual animals received a single score based on the most severe arrhythmia observed. 0 indicates no arrhythmia. 1 indicates 1–2 premature ventricular contractions (PVCs) followed by normal sinus rhythm of at least 2 beats. 2 indicates bigeminy (1 PVC followed by one normal sinus beat, repeating for 4 or more continuous cycles) or salvo (3–5 PVCs in a row). 3 indicates non-sustained ventricular

tachycardia (nsVT) defined as 6 or more PVCs in a row lasting less than 30 s. 4 indicates sustained VT (>30 s) or Torsades de Pointes.

## Statistics

Student's t-test was used for comparisons of just two samples. Data with more than two groups were analyzed by one-way ANOVA using the Tukey's post hoc test to compare all conditions or the Dunnet's post-test when comparing to a single control group. Data with multiple variables was analyzed by two-way ANOVA. All statistical analyses were carried out using Prism 9.

## Acknowledgements

The authors would like to thank Kevin Wright PhD for crucial comments on the manuscript and Ryan Gardner PhD for critical insight during the early stages and development of this project. We would also like to thank Tammi Howard for early technical assistance on this project. This work was supported by NIH F31HL152490 (MRB), AHA 20PRE35210768 (MRB), the Steinberg Endowment for Graduate Education (MRB), National Institutes of Health UL1GM118964 (SS), and NIH R01 HL093056 (BAH).

## Additional information

### Funding

| Funder | Grant reference number | Author |
|---|---|---|
| National Heart, Lung, and Blood Institute | HL093056 | Beth A Habecker |
| National Heart, Lung, and Blood Institute | F31HL152490 | Matthew R Blake |
| American Heart Association | 20PRE35210768 | Matthew R Blake |
| National Institutes of Health | UL1GM11864 | Shanice Sueda |

The funders had no role in study design, data collection and interpretation, or the decision to submit the work for publication.

### Author contributions

Matthew R Blake, Conceptualization, Data curation, Formal analysis, Funding acquisition, Investigation, Methodology, Project administration, Supervision, Validation, Visualization, Writing – original draft, Writing – review and editing; Diana C Parrish, Data curation, Investigation, Methodology, Validation; Melanie A Staffenson, Data curation, Investigation, Methodology; Shanice Sueda, Data curation, Formal analysis; William R Woodward, Conceptualization, Data curation, Formal analysis, Methodology, Supervision, Validation; Beth A Habecker, Conceptualization, Data curation, Formal analysis, Funding acquisition, Methodology, Project administration, Supervision, Writing – original draft, Writing – review and editing

### Author ORCIDs

Matthew R Blake ⓘ http://orcid.org/0000-0003-0584-7684
Beth A Habecker ⓘ http://orcid.org/0000-0002-4658-8730

### Ethics

All procedures were approved by the OHSU Institutional Animal Care and Use Committee (IACUC# TR01_IP00001366) and comply with the Guide for the Care and Use of Laboratory Animals published by the National Academies Press (8th edition). All myocardial infarction surgeries were performed under anesthesia induced with 4% isoflurane and maintained with 2% isoflurane. After surgery, animals were treated with buprenorphine and meloxicam to minimize suffering and discomfort.

### Decision letter and Author response

Decision letter https://doi.org/10.7554/eLife.78387.sa1

Author response https://doi.org/10.7554/eLife.78387.sa2

## Additional files

### Supplementary files
• Transparent reporting form
• Source data 1. Western blots.
• Supplementary file 1. Supplemental data – additional controls and troubleshooting for siRNA studies.

### Data availability
All data generated during this study are included in the manuscript and supporting files. Source data for annotated Western blot images from figure 1, 4, 5, 6 and S1, S2, S3 have been made available in the zipped western blot source data folder. Images are not cropped and they are labeled such that the figure number and protein blotted in each image are in the file name.

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
