## [Editor Report]

This is an extremely well performed study that advances the knowledge in this area and will foster further work. It also showcases how basic neuroscientific work can be applied to complex pathophysiological problems.

---

## [Decision Letter]

**Decision letter after peer review:**

Thank you for submitting your article "Chondroitin Sulfate Proteoglycan 4,6 sulfation regulates sympathetic nerve regeneration after myocardial infarction" for consideration by *eLife*. Your article has been reviewed by 4 peer reviewers, including Kalyanam Shivkumar as Reviewing Editor and Reviewer #1, and the evaluation has been overseen by Mone Zaidi as the Senior Editor. The following individuals involved in the review of your submission have agreed to reveal their identity: Hanjun Wang (Reviewer #2); Pradeep S Rajendran (Reviewer #4).

Summary of reviews:

Intramyocardial sympathetic nerve remodeling following myocardial infarction (MI) is associated with lethal ventricular arrhythmias and progression to heart failure. After an MI, there is sympathetic denervation of the infarct scar and hyperinnervation of the infarct border zone. This heterogeneity of innervation has been shown to contribute to electrical disorder and arrhythmogenesis. One mechanism for the lack of nerve regeneration in the infarct scar following MI has previously been shown to be the presence of chondroitin sulfate proteoglycans (CSPGs). In the present manuscript by Blake et al., the investigators show that MI leads to increased sulfation of CSPGs in the cardiac scar. The investigators subsequently demonstrate that reducing sulfation with a sulfatase, arylsulfatase B (ARSB), promotes sympathetic neurite growth in vitro and ex vivo in a co-culture system. Furthermore, molecular studies showed that there was an alteration in the expression of enzymes involved in the sulfation of CSPGs, specifically increased expression of a sulfotransferase, Chst15. Knockdown of the gene for Chst15 with a silencing RNA restored sympathetic innervation of the infarct scar, as shown by immunohistochemistry, and also reduced post-MI arrhythmias induced by isoproterenol in vivo, as shown by ECG.

Collectively, the reviewers were impressed with the depth and scope of the experiments presented here. Specifically, the manuscript's technical strengths were its logical progression from the authors' previous work on CSPGs in infarct scar, its finding of a novel CSPG sulfation pathway in MI, and its complementary mix of functional, histological and molecular studies to support its conclusions. The main weaknesses brought up by reviewers included a lack of complete rigor in some minor aspects of experimental design and a lack of an electrophysiological mechanism as to how the RNAi treatment led to decreased arrhythmic susceptibility.

MI with its subsequent ventricular arrhythmias is responsible for significant morbidity and mortality and represents an area of great therapeutic need. This work is important in highlighting the role of neural-myocardial interactions after MI and offering a potential pathway to target in preventing post-MI sudden cardiac death.

Essential revisions:

The paper needs to be edited for clarity and revised based on the input provided by the reviewers. No new experiments were sought by the reviewers.

*Reviewer #1 (Recommendations for the authors):*

1. Would consider treating scar tissue explants only with ARSB rather than the entire co-culture.

2. Would consider characterizing surviving myocardium and immune/inflammatory cells within the scar, as CSPG has been shown to be pro-inflammatory after neural injury (https://doi.org/10.1016/j.matbio.2018.04.010).

– If the siRNA treatment causes changes in scar/surviving myocardium structure, inflammatory milieu, or even vascularization of the scar, these would all be more plausible functional pathways for the observed reduction of PVCs than the presence of more nerves within an electrically and mechanically dead zone.

*Reviewer #2 (Recommendations for the authors):*

The primary objective of this study was to evaluate the potential contribution of chondroitin sulfate proteoglycan (CSPG) 4, 6 sulfation in the regulation of sympathetic nerve regeneration after ischemia-reperfusion (I/R). The authors found that I/R-induced myocardial injury led to increased 4, 6 sulfation of CSPGs in mouse cardiac scar, which was associated with upregulated CHST15 (4S dependent 6-sulfotransferase) and downregulated 4-sulfatase enzyme Arylsulfatase-B (ARSB). These molecular data suggest a mechanism for the production and maintenance of sulfated CSPGs in the cardiac scar tissue. Furthermore, the authors attempted to alter tandem sulfated 4s, 6s CS-GAGs in vivo by intravenous injection of Chst15 siRNA at the early recovery time window of I/R (i.e. before scar formation). They found that this treatment restored sympathetic reinnervation in the scar tissue and reduced cardiac arrhythmia induced by isoproterenol. Overall, this study is a logical follow-up to previous work from this group that had demonstrated that sympathetic nerves grow back through undamaged myocardium but do not enter the scar due to the presence of CSPGs. The paper presents a complementary mix of function, histological and molecular studies to provide mechanistic insights into disease progression. Results are well presented, between the primary and supplementary files, and the data support the conclusions.

*Reviewer #3 (Recommendations for the authors):*

– Figure 1B-D, could IHC be used to support these findings?

– Figure 3B – since the piece of the myocardium is just at the edge of the image, it should be highlighted with an arrow, or the image enlarge to show more of the myocardium. The outgrowth is also hard to see. Consider using arrows, or fixing/staining to improve visibility. This aspect of the data is not as convincing.

– Figure 4 – this would be strengthened by the addition of expression data from qPCR.

– Figure 7 – how can one exclude the potential role of systemic Chst15 targeting by the siRNA?

*Reviewer #4 (Recommendations for the authors):*

– In the introduction, a comparison is made between β-adrenergic blockade and sympathectomy to restore sympathetic innervation by modulating chondroitin sulfate proteoglycans. However, the mechanisms through which these approaches reduce arrhythmias are completely different and the comparison is somewhat confusing.

– While the arrhythmia susceptibility data is compelling, it would be useful to have cardiac electrophysiology data (e.g., optical mapping) in the presence of sympathetic nerve stimulation to show that the regenerated nerves are functional. In addition, cardiac electrophysiology data would help show where the ectopy is being generated from in the non-targeting control vs the siChst15 animals (i.e., scar border zone vs remote area).

---

## [Author Response]

Essential revisions:The paper needs to be edited for clarity and revised based on the input provided by the reviewers. No new experiments were sought by the reviewers.

We appreciate the comments from reviewers and the clarity of instruction from the editor.

Reviewer #1 (Recommendations for the authors):1. Would consider treating scar tissue explants only with ARSB rather than the entire co-culture.

Two different controls in our study suggest ARSB is acting on CSPGs produced in the cardiac scar rather than ganglionic CSPGs. First, ganglia co-cultured with control heart tissue were treated with ARSB and exhibited no difference in growth compared to ganglia grown with control heart tissue in the absence of ARSB. Second, ARSB did not change the growth of axons extending away from heart tissue explants where they are not in contact with cardiac CSPGs. If ganglionic/glial CSPGs were affecting outgrowth then ARSB should have increased axon growth in those conditions. We have clarified this in the text lines 130-134.

2. Would consider characterizing surviving myocardium and immune/inflammatory cells within the scar, as CSPG has been shown to be pro-inflammatory after neural injury (https://doi.org/10.1016/j.matbio.2018.04.010).– If the siRNA treatment causes changes in scar/surviving myocardium structure, inflammatory milieu, or even vascularization of the scar, these would all be more plausible functional pathways for the observed reduction of PVCs than the presence of more nerves within an electrically and mechanically dead zone.

We revised the manuscript to include discussion of data addressing these issues that was recently accepted by *JACC: Basic Translational Sciences* (lines 249-253 in the “clean” version of the revised text). We used therapeutics targeting the CSPG receptor PTP σ to restore innervation to the cardiac scar, and observed a shift in the immune response from pro-inflammatory toward reparative phenotypes. Therapeutics that caused a greater change in the immune response also decreased infarct size. This study targets CSPGs in the scar rather than their receptors on neurons, but it might similarly affect the immune response. Vascularization was unaltered by treatments that restored innervation.

Note: We do not yet have a DOI for the accepted JACC-BTS manuscript, but we expect it will be posted to their preprint server very soon so that we can add the DOI to the reference.

Reviewer #3 (Recommendations for the authors):– Figure 1B-D, could IHC be used to support these findings?

We previously used the pan-CSPG antibody CS56 to show that CSPGs are selectively enriched in the cardiac scar (Gardner 2013 reference). Unfortunately, the antibodies validated for identifying/quantifying sulfation by western blot do not work for IHC, and current methods for identifying sulfation require removing the GAG side chains from the core protein prior to analysis.

– Figure 3B – since the piece of the myocardium is just at the edge of the image, it should be highlighted with an arrow, or the image enlarge to show more of the myocardium. The outgrowth is also hard to see. Consider using arrows, or fixing/staining to improve visibility. This aspect of the data is not as convincing.

We revised this figure to enhance contrast and better label the myocardium.

– Figure 4 – this would be strengthened by the addition of expression data from qPCR.

We do not have RNA from these hearts as the tissue was all used for protein analysis. We are unclear what new information would be provided by examining gene expression.

– Figure 7 – how can one exclude the potential role of systemic Chst15 targeting by the siRNA?

We cannot exclude a potential role of systemic Chst15 targeting by the siRNA, but it is unclear how systemic effects of Chst15 might contribute to nerve regeneration into the cardiac scar. We have included this limitation (see lines 300-302 in the “clean” version of the revised text).

Reviewer #4 (Recommendations for the authors):– In the introduction, a comparison is made between β-adrenergic blockade and sympathectomy to restore sympathetic innervation by modulating chondroitin sulfate proteoglycans. However, the mechanisms through which these approaches reduce arrhythmias are completely different and the comparison is somewhat confusing.

This is a good point and we have revised the introduction to remove this comparison. Lines 64-66 in previous version deleted. This is easiest to see in the track changes document.

– While the arrhythmia susceptibility data is compelling, it would be useful to have cardiac electrophysiology data (e.g., optical mapping) in the presence of sympathetic nerve stimulation to show that the regenerated nerves are functional. In addition, cardiac electrophysiology data would help show where the ectopy is being generated from in the non-targeting control vs the siChst15 animals (i.e., scar border zone vs remote area).

We agree that mapping data would be very interesting and would shed light on the mechanisms of arrhythmia suppression. However, that was not the purpose of this study. We hope to do mapping studies in the future to address the issues you identified.